# Neural Tangent Kernel:
# Convergence and Generalization in Neural Networks

**Arthur Jacot**
École Polytechnique Fédérale de Lausanne
`arthur.jacot@netopera.net`

**Franck Gabriel**
Imperial College London and École Polytechnique Fédérale de Lausanne
`franckrgabriel@gmail.com`

**Clément Hongler**
École Polytechnique Fédérale de Lausanne
`clement.hongler@gmail.com`

## Abstract

At initialization, artificial neural networks (ANNs) are equivalent to Gaussian processes in the infinite-width limit (12; 9), thus connecting them to kernel methods. We prove that the evolution of an ANN during training can also be described by a kernel: during gradient descent on the parameters of an ANN, the network function $f_\theta$ (which maps input vectors to output vectors) follows the kernel gradient of the functional cost (which is convex, in contrast to the parameter cost) w.r.t. a new kernel: the Neural Tangent Kernel (NTK). This kernel is central to describe the generalization features of ANNs. While the NTK is random at initialization and varies during training, in the infinite-width limit it converges to an explicit limiting kernel and it stays constant during training. This makes it possible to study the training of ANNs in function space instead of parameter space. Convergence of the training can then be related to the positive-definiteness of the limiting NTK.

We then focus on the setting of least-squares regression and show that in the infinite-width limit, the network function $f_\theta$ follows a linear differential equation during training. The convergence is fastest along the largest kernel principal components of the input data with respect to the NTK, hence suggesting a theoretical motivation for early stopping.

Finally we study the NTK numerically, observe its behavior for wide networks, and compare it to the infinite-width limit.

## 1 Introduction

Artificial neural networks (ANNs) have achieved impressive results in numerous areas of machine learning. While it has long been known that ANNs can approximate any function with sufficiently many hidden neurons (7; 10), it is not known what the optimization of ANNs converges to. Indeed the loss surface of neural networks optimization problems is highly non-convex: it has a high number of saddle points which may slow down the convergence (4). A number of results (3; 13; 14) suggest that for wide enough networks, there are very few "bad" local minima, i.e. local minima with much higher cost than the global minimum. More recently, the investigation of the geometry of the loss landscape at initialization has been the subject of a precise study (8). The analysis of the dynamics

of training in the large-width limit for shallow networks has seen recent progress as well (11). To the best of the authors knowledge, the dynamics of deep networks has however remained an open problem until the present paper: see the contributions section below.

A particularly mysterious feature of ANNs is their good generalization properties in spite of their usual over-parametrization (16). It seems paradoxical that a reasonably large neural network can fit random labels, while still obtaining good test accuracy when trained on real data (19). It can be noted that in this case, kernel methods have the same properties (1).

In the infinite-width limit, ANNs have a Gaussian distribution described by a kernel (12; 9). These kernels are used in Bayesian inference or Support Vector Machines, yielding results comparable to ANNs trained with gradient descent (9; 2). We will see that in the same limit, the behavior of ANNs during training is described by a related kernel, which we call the neural tangent network (NTK).

## 1.1 Contribution

We study the network function $f_\theta$ of an ANN, which maps an input vector to an output vector, where $\theta$ is the vector of the parameters of the ANN. In the limit as the widths of the hidden layers tend to infinity, the network function at initialization, $f_\theta$ converges to a Gaussian distribution (12; 9).

In this paper, we investigate fully connected networks in this infinite-width limit, and describe the dynamics of the network function $f_\theta$ during training:

- During gradient descent, we show that the dynamics of $f_\theta$ follows that of the so-called *kernel gradient descent* in function space with respect to a limiting kernel, which only depends on the depth of the network, the choice of nonlinearity and the initialization variance.

- The convergence properties of ANNs during training can then be related to the positive-definiteness of the infinite-width limit NTK. The values of the network function $f_\theta$ outside the training set is described by the NTK, which is crucial to understand how ANN generalize.

- For a least-squares regression loss, the network function $f_\theta$ follows a linear differential equation in the infinite-width limit, and the eigenfunctions of the Jacobian are the kernel principal components of the input data. This shows a direct connection to kernel methods and motivates the use of early stopping to reduce overfitting in the training of ANNs.

- Finally we investigate these theoretical results numerically for an artificial dataset (of points on the unit circle) and for the MNIST dataset. In particular we observe that the behavior of wide ANNs is close to the theoretical limit.

## 2 Neural networks

In this article, we consider fully-connected ANNs with layers numbered from $0$ (input) to $L$ (output), each containing $n_0, \ldots, n_L$ neurons, and with a Lipschitz, twice differentiable nonlinearity function $\sigma : \mathbb{R} \to \mathbb{R}$, with bounded second derivative [1].

This paper focuses on the ANN *realization function* $F^{(L)} : \mathbb{R}^P \to \mathcal{F}$, mapping parameters $\theta$ to functions $f_\theta$ in a space $\mathcal{F}$. The dimension of the parameter space is $P = \sum_{\ell=0}^{L-1}(n_\ell + 1)n_{\ell+1}$: the parameters consist of the connection matrices $W^{(\ell)} \in \mathbb{R}^{n_\ell \times n_{\ell+1}}$ and bias vectors $b^{(\ell)} \in \mathbb{R}^{n_{\ell+1}}$ for $\ell = 0, ..., L-1$. In our setup, the parameters are initialized as iid Gaussians $\mathcal{N}(0, 1)$.

For a fixed distribution $p^{in}$ on the input space $\mathbb{R}^{n_0}$, the function space $\mathcal{F}$ is defined as $\{f : \mathbb{R}^{n_0} \to \mathbb{R}^{n_L}\}$. On this space, we consider the seminorm $|| \cdot ||_{p^{in}}$, defined in terms of the bilinear form

$$\langle f, g \rangle_{p^{in}} = \mathbb{E}_{x \sim p^{in}} \left[ f(x)^T g(x) \right].$$

In this paper, we assume that the input distribution $p^{in}$ is the empirical distribution on a finite dataset $x_1, ..., x_N$, i.e the sum of Dirac measures $\frac{1}{N} \sum_{i=0}^{N} \delta_{x_i}$.

We define the network function by $f_\theta(x) := \tilde{\alpha}^{(L)}(x;\theta)$, where the functions $\tilde{\alpha}^{(\ell)}(\cdot;\theta) : \mathbb{R}^{n_0} \to \mathbb{R}^{n_\ell}$ (called *preactivations*) and $\alpha^{(\ell)}(\cdot;\theta) : \mathbb{R}^{n_0} \to \mathbb{R}^{n_\ell}$ (called *activations*) are defined from the 0-th to the $L$-th layer by:

$$\alpha^{(0)}(x;\theta) = x$$
$$\tilde{\alpha}^{(\ell+1)}(x;\theta) = \frac{1}{\sqrt{n_\ell}} W^{(\ell)}\alpha^{(\ell)}(x;\theta) + \beta b^{(\ell)}$$
$$\alpha^{(\ell)}(x;\theta) = \sigma(\tilde{\alpha}^{(\ell)}(x;\theta)),$$

where the nonlinearity $\sigma$ is applied entrywise. The scalar $\beta > 0$ is a parameter which allows us to tune the influence of the bias on the training.

**Remark 1.** *Our definition of the realization function $F^{(L)}$ slightly differs from the classical one. Usually, the factors $\frac{1}{\sqrt{n_\ell}}$ and the parameter $\beta$ are absent and the parameters are initialized using what is sometimes called LeCun initialization, taking $W_{ij}^{(\ell)} \sim \mathcal{N}(0,\frac{1}{n_\ell})$ and $b_j^{(\ell)} \sim \mathcal{N}(0,1)$ (or sometimes $b_j^{(\ell)} = 0$) to compensate. While the set of representable functions $F^{(L)}(\mathbb{R}^P)$ is the same for both parametrizations (with or without the factors $\frac{1}{\sqrt{n_\ell}}$ and $\beta$), the derivatives of the realization function with respect to the connections $\partial_{W_{ij}^{(\ell)}} F^{(L)}$ and bias $\partial_{b_j^{(\ell)}} F^{(L)}$ are scaled by $\frac{1}{\sqrt{n_\ell}}$ and $\beta$ respectively in comparison to the classical parametrization.*

*The factors $\frac{1}{\sqrt{n_\ell}}$ are key to obtaining a consistent asymptotic behavior of neural networks as the widths of the hidden layers $n_1, ..., n_{L-1}$ grow to infinity. However a side-effect of these factors is that they reduce greatly the influence of the connection weights during training when $n_\ell$ is large: the factor $\beta$ is introduced to balance the influence of the bias and connection weights. In our numerical experiments, we take $\beta = 0.1$ and use a learning rate of $1.0$, which is larger than usual, see Section 6. This gives a behaviour similar to that of a classical network of width $100$ with a learning rate of $0.01$.*

## 3 Kernel gradient

The training of an ANN consists in optimizing $f_\theta$ in the function space $\mathcal{F}$ with respect to a functional cost $C : \mathcal{F} \to \mathbb{R}$, such as a regression or cross-entropy cost. Even for a convex functional cost $C$, the composite cost $C \circ F^{(L)} : \mathbb{R}^P \to \mathbb{R}$ is in general highly non-convex (3). We will show that during training, the network function $f_\theta$ follows a descent along the kernel gradient with respect to the Neural Tangent Kernel (NTK) which we introduce in Section 4. This makes it possible to study the training of ANNs in the function space $\mathcal{F}$, on which the cost $C$ is convex.

A *multi-dimensional kernel* $K$ is a function $\mathbb{R}^{n_0} \times \mathbb{R}^{n_0} \to \mathbb{R}^{n_L \times n_L}$, which maps any pair $(x, x')$ to an $n_L \times n_L$-matrix such that $K(x,x') = K(x',x)^T$ (equivalently $K$ is a symmetric tensor in $\mathcal{F} \otimes \mathcal{F}$). Such a kernel defines a bilinear map on $\mathcal{F}$, taking the expectation over independent $x, x' \sim p^{in}$:

$$\langle f, g\rangle_K := \mathbb{E}_{x,x'\sim p^{in}}\left[ f(x)^T K(x,x')g(x') \right].$$

The kernel $K$ is *positive definite with respect to* $||\cdot||_{p^{in}}$ if $||f||_{p^{in}} > 0 \implies ||f||_K > 0$.

We denote by $\mathcal{F}^*$ the dual of $\mathcal{F}$ with respect to $p^{in}$, i.e. the set of linear forms $\mu : \mathcal{F} \to \mathbb{R}$ of the form $\mu = \langle d, \cdot\rangle_{p^{in}}$ for some $d \in \mathcal{F}$. Two elements of $\mathcal{F}$ define the same linear form if and only if they are equal on the data. The constructions in the paper do not depend on the element $d \in \mathcal{F}$ chosen in order to represent $\mu$ as $\langle d, \cdot\rangle_{p^{in}}$. Using the fact that the partial application of the kernel $K_{i,\cdot}(x,\cdot)$ is a function in $\mathcal{F}$, we can define a map $\Phi_K : \mathcal{F}^* \to \mathcal{F}$ mapping a dual element $\mu = \langle d, \cdot\rangle_{p^{in}}$ to the function $f_\mu = \Phi_K(\mu)$ with values:

$$f_{\mu,i}(x) = \mu K_{i,\cdot}(x,\cdot) = \langle d, K_{i,\cdot}(x,\cdot)\rangle_{p^{in}}.$$

For our setup, which is that of a finite dataset $x_1, \ldots, x_n \in \mathbb{R}^{n_0}$, the cost functional $C$ only depends on the values of $f \in \mathcal{F}$ at the data points. As a result, the (functional) derivative of the cost $C$ at a point $f_0 \in \mathcal{F}$ can be viewed as an element of $\mathcal{F}^*$, which we write $\partial_f^{in} C|_{f_0}$. We denote by $d|_{f_0} \in \mathcal{F}$, a corresponding dual element, such that $\partial_f^{in} C|_{f_0} = \langle d|_{f_0}, \cdot\rangle_{p^{in}}$.

The *kernel gradient* $\nabla_K C|_{f_0} \in \mathcal{F}$ is defined as $\Phi_K \left( \partial_f^{in} C|_{f_0} \right)$. In contrast to $\partial_f^{in} C$ which is only defined on the dataset, the kernel gradient generalizes to values $x$ outside the dataset thanks to the kernel $K$:

$$\nabla_K C|_{f_0}(x) = \frac{1}{N} \sum_{j=1}^{N} K(x, x_j) d|_{f_0}(x_j).$$

A time-dependent function $f(t)$ follows the *kernel gradient descent with respect to $K$* if it satisfies the differential equation

$$\partial_t f(t) = -\nabla_K C|_{f(t)}.$$

During kernel gradient descent, the cost $C(f(t))$ evolves as

$$\partial_t C|_{f(t)} = - \left\langle d|_{f(t)}, \nabla_K C|_{f(t)} \right\rangle_{p^{in}} = - \left\| d|_{f(t)} \right\|_K^2.$$

Convergence to a critical point of $C$ is hence guaranteed if the kernel $K$ is positive definite with respect to $\| \cdot \|_{p^{in}}$: the cost is then strictly decreasing except at points such that $\|d|_{f(t)}\|_{p^{in}} = 0$. If the cost is convex and bounded from below, the function $f(t)$ therefore converges to a global minimum as $t \to \infty$.

## 3.1 Random functions approximation

As a starting point to understand the convergence of ANN gradient descent to kernel gradient descent in the infinite-width limit, we introduce a simple model, inspired by the approach of (15).

A kernel $K$ can be approximated by a choice of $P$ random functions $f^{(p)}$ sampled independently from any distribution on $\mathcal{F}$ whose (non-centered) covariance is given by the kernel $K$:

$$\mathbb{E}[f_k^{(p)}(x) f_{k'}^{(p)}(x')] = K_{kk'}(x, x').$$

These functions define a random linear parametrization $F^{lin} : \mathbb{R}^P \to \mathcal{F}$

$$\theta \mapsto f_\theta^{lin} = \frac{1}{\sqrt{P}} \sum_{p=1}^{P} \theta_p f^{(p)}.$$

The partial derivatives of the parametrization are given by

$$\partial_{\theta_p} F^{lin}(\theta) = \frac{1}{\sqrt{P}} f^{(p)}.$$

Optimizing the cost $C \circ F^{lin}$ through gradient descent, the parameters follow the ODE:

$$\partial_t \theta_p(t) = -\partial_{\theta_p}(C \circ F^{lin})(\theta(t)) = -\frac{1}{\sqrt{P}} \partial_f^{in} C|_{f_{\theta(t)}^{lin}} f^{(p)} = -\frac{1}{\sqrt{P}} \left\langle d|_{f_{\theta(t)}^{lin}}, f^{(p)} \right\rangle_{p^{in}}.$$

As a result the function $f_{\theta(t)}^{lin}$ evolves according to

$$\partial_t f_{\theta(t)}^{lin} = \frac{1}{\sqrt{P}} \sum_{p=1}^{P} \partial_t \theta_p(t) f^{(p)} = -\frac{1}{P} \sum_{p=1}^{P} \left\langle d|_{f_{\theta(t)}^{lin}}, f^{(p)} \right\rangle_{p^{in}} f^{(p)},$$

where the right-hand side is equal to the kernel gradient $-\nabla_{\tilde{K}} C$ with respect to the *tangent kernel*

$$\tilde{K} = \sum_{p=1}^{P} \partial_{\theta_p} F^{lin}(\theta) \otimes \partial_{\theta_p} F^{lin}(\theta) = \frac{1}{P} \sum_{p=1}^{P} f^{(p)} \otimes f^{(p)}.$$

This is a random $n_L$-dimensional kernel with values $\tilde{K}_{ii'}(x, x') = \frac{1}{P} \sum_{p=1}^{P} f_i^{(p)}(x) f_{i'}^{(p)}(x')$.

Performing gradient descent on the cost $C \circ F^{lin}$ is therefore equivalent to performing kernel gradient descent with the tangent kernel $\tilde{K}$ in the function space. In the limit as $P \to \infty$, by the law of large numbers, the (random) tangent kernel $\tilde{K}$ tends to the fixed kernel $K$, which makes this method an approximation of kernel gradient descent with respect to the limiting kernel $K$.

# 4 Neural tangent kernel

For ANNs trained using gradient descent on the composition $C \circ F^{(L)}$, the situation is very similar to that studied in the Section 3.1. During training, the network function $f_\theta$ evolves along the (negative) kernel gradient

$$\partial_t f_{\theta(t)} = -\nabla_{\Theta^{(L)}} C|_{f_{\theta(t)}}$$

with respect to the *neural tangent kernel* (NTK)

$$\Theta^{(L)}(\theta) = \sum_{p=1}^{P} \partial_{\theta_p} F^{(L)}(\theta) \otimes \partial_{\theta_p} F^{(L)}(\theta).$$

However, in contrast to $F^{lin}$, the realization function $F^{(L)}$ of ANNs is not linear. As a consequence, the derivatives $\partial_{\theta_p} F^{(L)}(\theta)$ and the neural tangent kernel depend on the parameters $\theta$. The NTK is therefore random at initialization and varies during training, which makes the analysis of the convergence of $f_\theta$ more delicate.

In the next subsections, we show that, in the infinite-width limit, the NTK becomes deterministic at initialization and stays constant during training. Since $f_\theta$ at initialization is Gaussian in the limit, the asymptotic behavior of $f_\theta$ during training can be explicited in the function space $\mathcal{F}$.

## 4.1 Initialization

As observed in (12; 9), the output functions $f_{\theta,i}$ for $i = 1, ..., n_L$ tend to iid Gaussian processes in the infinite-width limit (a proof in our setup is given in the appendix):

**Proposition 1.** *For a network of depth $L$ at initialization, with a Lipschitz nonlinearity $\sigma$, and in the limit as $n_1, ..., n_{L-1} \to \infty$, the output functions $f_{\theta,k}$, for $k = 1, ..., n_L$, tend (in law) to iid centered Gaussian processes of covariance $\Sigma^{(L)}$, where $\Sigma^{(L)}$ is defined recursively by:*

$$\Sigma^{(1)}(x, x') = \frac{1}{n_0} x^T x' + \beta^2$$
$$\Sigma^{(L+1)}(x, x') = \mathbb{E}_{f \sim \mathcal{N}(0, \Sigma^{(L)})} [\sigma(f(x))\sigma(f(x'))] + \beta^2,$$

*taking the expectation with respect to a centered Gaussian process $f$ of covariance $\Sigma^{(L)}$.*

**Remark 2.** *Strictly speaking, the existence of a suitable Gaussian measure with covariance $\Sigma^{(L)}$ is not needed: we only deal with the values of $f$ at $x, x'$ (the joint measure on $f(x), f(x')$ is simply a Gaussian vector in 2D). For the same reasons, in the proof of Proposition 1 and Theorem 1, we will freely speak of Gaussian processes without discussing their existence.*

The first key result of our paper (proven in the appendix) is the following: in the same limit, the Neural Tangent Kernel (NTK) converges in probability to an explicit deterministic limit.

**Theorem 1.** *For a network of depth $L$ at initialization, with a Lipschitz nonlinearity $\sigma$, and in the limit as the layers width $n_1, ..., n_{L-1} \to \infty$, the NTK $\Theta^{(L)}$ converges in probability to a deterministic limiting kernel:*

$$\Theta^{(L)} \to \Theta_\infty^{(L)} \otimes Id_{n_L}.$$

*The scalar kernel $\Theta_\infty^{(L)} : \mathbb{R}^{n_0} \times \mathbb{R}^{n_0} \to \mathbb{R}$ is defined recursively by*

$$\Theta_\infty^{(1)}(x, x') = \Sigma^{(1)}(x, x')$$
$$\Theta_\infty^{(L+1)}(x, x') = \Theta_\infty^{(L)}(x, x')\dot{\Sigma}^{(L+1)}(x, x') + \Sigma^{(L+1)}(x, x'),$$

*where*

$$\dot{\Sigma}^{(L+1)}(x, x') = \mathbb{E}_{f \sim \mathcal{N}(0, \Sigma^{(L)})} [\dot{\sigma}(f(x))\dot{\sigma}(f(x'))],$$

*taking the expectation with respect to a centered Gaussian process $f$ of covariance $\Sigma^{(L)}$, and where $\dot{\sigma}$ denotes the derivative of $\sigma$.*

**Remark 3.** *By Rademacher's theorem, $\dot{\sigma}$ is defined everywhere, except perhaps on a set of zero Lebesgue measure.*

Note that the limiting $\Theta_\infty^{(L)}$ only depends on the choice of $\sigma$, the depth of the network and the variance of the parameters at initialization (which is equal to 1 in our setting).

## 4.2 Training

Our second key result is that the NTK stays asymptotically constant during training. This applies for a slightly more general definition of training: the parameters are updated according to a training direction $d_t \in \mathcal{F}$:

$$\partial_t \theta_p(t) = \left\langle \partial_{\theta_p} F^{(L)}(\theta(t)), d_t \right\rangle_{p^{in}}.$$

In the case of gradient descent, $d_t = -d|_{f_{\theta(t)}}$ (see Section 3), but the direction may depend on another network, as is the case for e.g. Generative Adversarial Networks (6). We only assume that the integral $\int_0^T \|d_t\|_{p^{in}} dt$ stays stochastically bounded as the width tends to infinity, which is verified for e.g. least-squares regression, see Section 5.

**Theorem 2.** *Assume that $\sigma$ is a Lipschitz, twice differentiable nonlinearity function, with bounded second derivative. For any $T$ such that the integral $\int_0^T \|d_t\|_{p^{in}} dt$ stays stochastically bounded, as $n_1, ..., n_{L-1} \to \infty$, we have, uniformly for $t \in [0, T]$,*

$$\Theta^{(L)}(t) \to \Theta_\infty^{(L)} \otimes Id_{n_L}.$$

*As a consequence, in this limit, the dynamics of $f_\theta$ is described by the differential equation*

$$\partial_t f_{\theta(t)} = \Phi_{\Theta_\infty^{(L)} \otimes Id_{n_L}} \left( \langle d_t, \cdot \rangle_{p^{in}} \right).$$

**Remark 4.** *As the proof of the theorem (in the appendix) shows, the variation during training of the individual activations in the hidden layers shrinks as their width grows. However their collective variation is significant, which allows the parameters of the lower layers to learn: in the formula of the limiting NTK $\Theta_\infty^{(L+1)}(x, x')$ in Theorem 1, the second summand $\Sigma^{(L+1)}$ represents the learning due to the last layer, while the first summand represents the learning performed by the lower layers.*

As discussed in Section 3, the convergence of kernel gradient descent to a critical point of the cost $C$ is guaranteed for positive definite kernels. The limiting NTK is positive definite if the span of the derivatives $\partial_{\theta_p} F^{(L)}$, $p = 1, ..., P$ becomes dense in $\mathcal{F}$ w.r.t. the $p^{in}$-norm as the width grows to infinity. It seems natural to postulate that the span of the preactivations of the last layer (which themselves appear in $\partial_{\theta_p} F^{(L)}$, corresponding to the connection weights of the last layer) becomes dense in $\mathcal{F}$, for a large family of measures $p^{in}$ and nonlinearities (see e.g. (7; 10) for classical theorems about ANNs and approximation).

## 5   Least-squares regression

Given a goal function $f^*$ and input distribution $p^{in}$, the least-squares regression cost is

$$C(f) = \frac{1}{2} \|f - f^*\|_{p^{in}}^2 = \frac{1}{2} \mathbb{E}_{x \sim p^{in}} \left[ \|f(x) - f^*(x)\|^2 \right].$$

Theorems 1 and 2 apply to an ANN trained on such a cost. Indeed the norm of the training direction $\|d(f)\|_{p^{in}} = \|f^* - f\|_{p^{in}}$ is strictly decreasing during training, bounding the integral. We are therefore interested in the behavior of a function $f_t$ during kernel gradient descent with a kernel $K$ (we are of course especially interested in the case $K = \Theta_\infty^{(L)} \otimes Id_{n_L}$):

$$\partial_t f_t = \Phi_K \left( \langle f^* - f, \cdot \rangle_{p^{in}} \right).$$

The solution of this differential equation can be expressed in terms of the map $\Pi : f \mapsto \Phi_K \left( \langle f, \cdot \rangle_{p^{in}} \right)$:

$$f_t = f^* + e^{-t\Pi}(f_0 - f^*)$$

where $e^{-t\Pi} = \sum_{k=0}^\infty \frac{(-t)^k}{k!} \Pi^k$ is the exponential of $-t\Pi$. If $\Pi$ can be diagonalized by eigenfunctions $f^{(i)}$ with eigenvalues $\lambda_i$, the exponential $e^{-t\Pi}$ has the same eigenfunctions with eigenvalues $e^{-t\lambda_i}$.

For a finite dataset $x_1, ..., x_N$ of size $N$, the map $\Pi$ takes the form

$$\Pi(f)_k(x) = \frac{1}{N} \sum_{i=1}^N \sum_{k'=1}^{n_L} f_{k'}(x_i) K_{kk'}(x_i, x).$$

The map $\Pi$ has at most $Nn_L$ positive eigenfunctions, and they are the kernel principal components $f^{(1)}, ..., f^{(Nn_L)}$ of the data with respect to to the kernel $K$ (17; 18). The corresponding eigenvalues $\lambda_i$ is the variance captured by the component.

Decomposing the difference $(f^* - f_0) = \Delta_f^0 + \Delta_f^1 + ... + \Delta_f^{Nn_L}$ along the eigenspaces of $\Pi$, the trajectory of the function $f_t$ reads

$$f_t = f^* + \Delta_f^0 + \sum_{i=1}^{Nn_L} e^{-t\lambda_i} \Delta_f^i,$$

where $\Delta_f^0$ is in the kernel (null-space) of $\Pi$ and $\Delta_f^i \propto f^{(i)}$.

The above decomposition can be seen as a motivation for the use of early stopping. The convergence is indeed faster along the eigenspaces corresponding to larger eigenvalues $\lambda_i$. Early stopping hence focuses the convergence on the most relevant kernel principal components, while avoiding to fit the ones in eigenspaces with lower eigenvalues (such directions are typically the 'noisier' ones: for instance, in the case of the RBF kernel, lower eigenvalues correspond to high frequency functions).

Note that by the linearity of the map $e^{-t\Pi}$, if $f_0$ is initialized with a Gaussian distribution (as is the case for ANNs in the infinite-width limit), then $f_t$ is Gaussian for all times $t$. Assuming that the kernel is positive definite on the data (implying that the $Nn_L \times Nn_L$ Gram marix $\tilde{K} = (K_{kk'}(x_i, x_j))_{ik,jk'}$ is invertible), as $t \to \infty$ limit, we get that $f_\infty = f^* + \Delta_f^0 = f_0 - \sum_i \Delta_f^i$ takes the form

$$f_{\infty,k}(x) = \kappa_{x,k}^T \tilde{K}^{-1} y^* + \left( f_0(x) - \kappa_{x,k}^T \tilde{K}^{-1} y_0 \right),$$

with the $Nn_l$-vectors $\kappa_{x,k}$, $y^*$ and $y_0$ given by

$$\kappa_{x,k} = (K_{kk'}(x, x_i))_{i,k'}$$
$$y^* = (f_k^*(x_i))_{i,k}$$
$$y_0 = (f_{0,k}(x_i))_{i,k}.$$

The first term, the mean, has an important statistical interpretation: it is the maximum-a-posteriori (MAP) estimate given a Gaussian prior on functions $f_k \sim \mathcal{N}(0, \Theta_\infty^{(L)})$ and the conditions $f_k(x_i) = f_k^*(x_i)$. Equivalently, it is equal to the kernel ridge regression (18) as the regularization goes to zero ($\lambda \to 0$). The second term is a centered Gaussian whose variance vanishes on the points of the dataset.

## 6 Numerical experiments

In the following numerical experiments, fully connected ANNs of various widths are compared to the theoretical infinite-width limit. We choose the size of the hidden layers to all be equal to the same value $n := n_1 = ... = n_{L-1}$ and we take the ReLU nonlinearity $\sigma(x) = \max(0, x)$.

In the first two experiments, we consider the case $n_0 = 2$. Moreover, the input elements are taken on the unit circle. This can be motivated by the structure of high-dimensional data, where the centered data points often have roughly the same norm [2].

In all experiments, we took $n_L = 1$ (note that by our results, a network with $n_L$ outputs behaves asymptotically like $n_L$ networks with scalar outputs trained independently). Finally, the value of the parameter $\beta$ is chosen as 0.1, see Remark 1.

### 6.1 Convergence of the NTK

The first experiment illustrates the convergence of the NTK $\Theta^{(L)}$ of a network of depth $L = 4$ for two different widths $n = 500, 10000$. The function $\Theta^{(4)}(x_0, x)$ is plotted for a fixed $x_0 = (1, 0)$ and $x = (cos(\gamma), sin(\gamma))$ on the unit circle in Figure 1. To observe the distribution of the NTK, 10 independent initializations are performed for both widths. The kernels are plotted at initialization

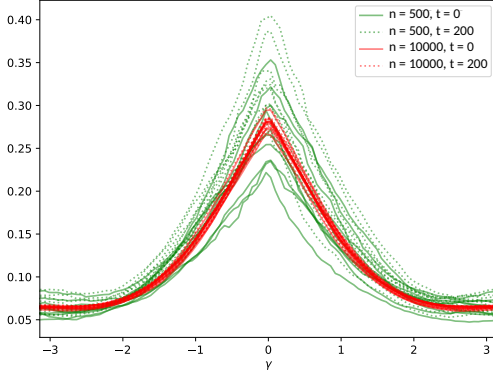 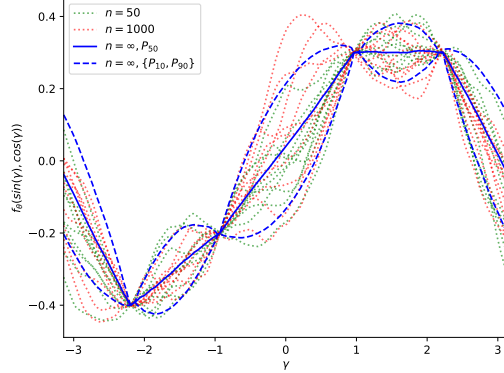

Figure 1: Convergence of the NTK to a fixed limit for two widths $n$ and two times $t$.

Figure 2: Networks function $f_\theta$ near convergence for two widths $n$ and 10th, 50th and 90th percentiles of the asymptotic Gaussian distribution.

$t = 0$ and then after 200 steps of gradient descent with learning rate 1.0 (i.e. at $t = 200$). We approximate the function $f^*(x) = x_1 x_2$ with a least-squares cost on random $\mathcal{N}(0, 1)$ inputs.

For the wider network, the NTK shows less variance and is smoother. It is interesting to note that the expectation of the NTK is very close for both networks widths. After 200 steps of training, we observe that the NTK tends to "inflate". As expected, this effect is much less apparent for the wider network ($n = 10000$) where the NTK stays almost fixed, than for the smaller network ($n = 500$).

### 6.2  Kernel regression

For a regression cost, the infinite-width limit network function $f_{\theta(t)}$ has a Gaussian distribution for all times $t$ and in particular at convergence $t \to \infty$ (see Section 5). We compared the theoretical Gaussian distribution at $t \to \infty$ to the distribution of the network function $f_{\theta(T)}$ of a finite-width network for a large time $T = 1000$. For two different widths $n = 50, 1000$ and for 10 random initializations each, a network is trained on a least-squares cost on 4 points of the unit circle for 1000 steps with learning rate 1.0 and then plotted in Figure 2.

We also approximated the kernels $\Theta_\infty^{(4)}$ and $\Sigma^{(4)}$ using a large-width network ($n = 10000$) and used them to calculate and plot the 10th, 50th and 90-th percentiles of the $t \to \infty$ limiting Gaussian distribution.

The distributions of the network functions are very similar for both widths: their mean and variance appear to be close to those of the limiting distribution $t \to \infty$. Even for relatively small widths ($n = 50$), the NTK gives a good indication of the distribution of $f_{\theta(t)}$ as $t \to \infty$.

### 6.3  Convergence along a principal component

We now illustrate our result on the MNIST dataset of handwritten digits made up of grayscale images of dimension $28 \times 28$, yielding a dimension of $n_0 = 784$.

We computed the first 3 principal components of a batch of $N = 512$ digits with respect to the NTK of a high-width network $n = 10000$ (giving an approximation of the limiting kernel) using a power iteration method. The respective eigenvalues are $\lambda_1 = 0.0457$, $\lambda_2 = 0.00108$ and $\lambda_3 = 0.00078$. The kernel PCA is non-centered, the first component is therefore almost equal to the constant function, which explains the large gap between the first and second eigenvalues[3]. The next two components are much more interesting as can be seen in Figure 3a, where the batch is plotted with $x$ and $y$ coordinates corresponding to the 2nd and 3rd components.

We have seen in Section 5 how the convergence of kernel gradient descent follows the kernel principal components. If the difference at initialization $f_0 - f^*$ is equal (or proportional) to one of the principal

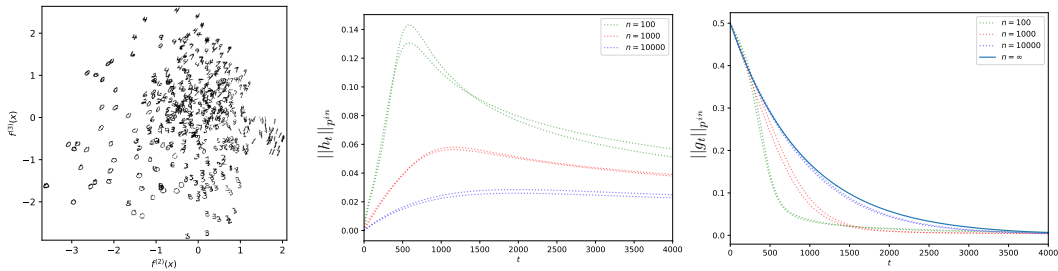

(a) The 2nd and 3rd principal components of MNIST.

(b) Deviation of the network function $f_\theta$ from the straight line.

(c) Convergence of $f_\theta$ along the 2nd principal component.

Figure 3

components $f^{(i)}$, then the function will converge along a straight line (in the function space) to $f^*$ at an exponential rate $e^{-\lambda_i t}$.

We tested whether ANNs of various widths $n = 100, 1000, 10000$ behave in a similar manner. We set the goal of the regression cost to $f^* = f_{\theta(0)} + 0.5 f^{(2)}$ and let the network converge. At each time step $t$, we decomposed the difference $f_{\theta(t)} - f^*$ into a component $g_t$ proportional to $f^{(2)}$ and another one $h_t$ orthogonal to $f^{(2)}$. In the infinite-width limit, the first component decays exponentially fast $||g_t||_{p^{in}} = 0.5 e^{-\lambda_2 t}$ while the second is null ($h_t = 0$), as the function converges along a straight line.

As expected, we see in Figure 3b that the wider the network, the less it deviates from the straight line (for each width $n$ we performed two independent trials). As the width grows, the trajectory along the 2nd principal component (shown in Figure 3c) converges to the theoretical limit shown in blue.

A surprising observation is that smaller networks appear to converge faster than wider ones. This may be explained by the inflation of the NTK observed in our first experiment. Indeed, multiplying the NTK by a factor $a$ is equivalent to multiplying the learning rate by the same factor. However, note that since the NTK of large-width network is more stable during training, larger learning rates can in principle be taken. One must hence be careful when comparing the convergence speed in terms of the number of steps (rather than in terms of the time $t$): both the inflation effect and the learning rate must be taken into account.

## 7 Conclusion

This paper introduces a new tool to study ANNs, the Neural Tangent Kernel (NTK), which describes the local dynamics of an ANN during gradient descent. This leads to a new connection between ANN training and kernel methods: in the infinite-width limit, an ANN can be described in the function space directly by the limit of the NTK, an explicit constant kernel $\Theta_\infty^{(L)}$, which only depends on its depth, nonlinearity and parameter initialization variance. More precisely, in this limit, ANN gradient descent is shown to be equivalent to a kernel gradient descent with respect to $\Theta_\infty^{(L)}$. The limit of the NTK is hence a powerful tool to understand the generalization properties of ANNs, and it allows one to study the influence of the depth and nonlinearity on the learning abilities of the network. The analysis of training using NTK allows one to relate convergence of ANN training with the positive-definiteness of the limiting NTK and leads to a characterization of the directions favored by early stopping methods.

## Acknowledgements

The authors thank K. Kytölä for many interesting discussions. The second author was supported by the ERC CG CRITICAL. The last author acknowledges support from the ERC SG Constamis, the NCCR SwissMAP, the Blavatnik Family Foundation and the Latsis Foundation.

## Footnotes

[1]While these smoothness assumptions greatly simplify the proofs of our results, they do not seem to be strictly needed for the results to hold true.

[2]The classical example is for data following a Gaussian distribution $\mathcal{N}(0, Id_{n_0})$: as the dimension $n_0$ grows, all data points have approximately the same norm $\sqrt{n_0}$.

[3]It can be observed numerically, that if we choose $\beta = 1.0$ instead of our recommended 0.1, the gap between the first and the second principal component is about ten times bigger, which makes training more difficult.

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
