[Supplementary Material]

# Neural Tangent Kernel:
# Convergence and Generalization in Neural Networks

**Arthur Jacot**
École Polytechnique Fédérale de Lausanne
`arthur.jacot@netopera.net`

**Franck Gabriel**
Imperial College London and École Polytechnique Fédérale de Lausanne
`franckrgabriel@gmail.com`

**Clément Hongler**
École Polytechnique Fédérale de Lausanne
`clement.hongler@gmail.com`

## A  Appendix

This appendix is dedicated to proving the key results of this paper, namely Proposition 1 and Theorems 1 and 2, which describe the asymptotics of neural networks at initialization and during training.

We study the limit of the NTK as $n_1, ..., n_{L-1} \to \infty$ sequentially, i.e. we first take $n_1 \to \infty$, then $n_2 \to \infty$, etc. This leads to much simpler proofs, but our results could in principle be strengthened to the more general setting when $\min(n_1, ..., n_{L-1}) \to \infty$.

A natural choice of convergence to study the NTK is with respect to the operator norm on kernels:

$$\|K\|_{op} = \max_{\|f\|_{p^{in}} \leq 1} \|f\|_K = \max_{\|f\|_{p^{in}} \leq 1} \sqrt{\mathbb{E}_{x,x'}[f(x)^T K(x,x') f(x')]},$$

where the expectation is taken over two independent $x, x' \sim p^{in}$. This norm depends on the input distribution $p^{in}$. In our setting, $p^{in}$ is taken to be the empirical measure of a finite dataset of distinct samples $x_1, ..., x_N$. As a result, the operator norm of $K$ is equal to the leading eigenvalue of the $Nn_L \times Nn_L$ Gram matrix $(K_{kk'}(x_i, x_j))_{k,k'<n_L, i,j<N}$. In our setting, convergence in operator norm is hence equivalent to pointwise convergence of $K$ on the dataset.

### A.1  Asymptotics at Initialization

It has already been observed (12; 9) that the output functions $f_{\theta,i}$ for $i = 1, ..., n_L$ tend to iid Gaussian processes in the infinite-width limit.

**Proposition 1.** *For a network of depth $L$ at initialization, with a Lipschitz nonlinearity $\sigma$, and in the limit as $n_1, ..., n_{L-1} \to \infty$ sequentially, the output functions $f_{\theta,k}$, for $k = 1, ..., n_L$, tend (in law) to iid centered Gaussian processes of covariance $\Sigma^{(L)}$, where $\Sigma^{(L)}$ is defined recursively by:*

$$\Sigma^{(1)}(x, x') = \frac{1}{n_0} x^T x' + \beta^2$$
$$\Sigma^{(L+1)}(x, x') = \mathbb{E}_f[\sigma(f(x))\sigma(f(x'))] + \beta^2,$$

*taking the expectation with respect to a centered Gaussian process $f$ of covariance $\Sigma^{(L)}$.*

*Proof.* We prove the result by induction. When $L = 1$, there are no hidden layers and $f_\theta$ is a random affine function of the form:

$$f_\theta(x) = \frac{1}{\sqrt{n_0}} W^{(0)} x + \beta b^{(0)}.$$

All output functions $f_{\theta,k}$ are hence independent and have covariance $\Sigma^{(1)}$ as needed.

The key to the induction step is to consider an $(L + 1)$-network as the following composition: an $L$-network $\mathbb{R}^{n_0} \to \mathbb{R}^{n_L}$ mapping the input to the pre-activations $\tilde{\alpha}_i^{(L)}$, followed by an elementwise application of the nonlinearity $\sigma$ and then a random affine map $\mathbb{R}^{n_L} \to \mathbb{R}^{n_{L+1}}$. The induction hypothesis gives that in the limit as sequentially $n_1, ..., n_{L-1} \to \infty$ the preactivations $\tilde{\alpha}_i^{(L)}$ tend to iid Gaussian processes with covariance $\Sigma^{(L)}$. The outputs

$$f_{\theta,i} = \frac{1}{\sqrt{n_L}} W_i^{(L)} \alpha^{(L)} + \beta b_i^{(L)}$$

conditioned on the values of $\alpha^{(L)}$ are iid centered Gaussians with covariance

$$\tilde{\Sigma}^{(L+1)}(x, x') = \frac{1}{n_L} \alpha^{(L)}(x; \theta)^T \alpha^{(L)}(x'; \theta) + \beta^2.$$

By the law of large numbers, as $n_L \to \infty$, this covariance tends in probability to the expectation

$$\tilde{\Sigma}^{(L+1)}(x, x') \to \Sigma^{(L+1)}(x, x') = \mathbb{E}_{f \sim \mathcal{N}(0,\Sigma^{(L)})}[\sigma(f(x))\sigma(f(x'))] + \beta^2.$$

In particular the covariance is deterministic and hence independent of $\alpha^{(L)}$. As a consequence, the conditioned and unconditioned distributions of $f_{\theta,i}$ are equal in the limit: they are iid centered Gaussian of covariance $\Sigma^{(L+1)}$. □

In the infinite-width limit, the neural tangent kernel, which is random at initialization, converges in probability to a deterministic limit.

**Theorem 1.** *For a network of depth $L$ at initialization, with a Lipschitz nonlinearity $\sigma$, and in the limit as the layers width $n_1, ..., n_{L-1} \to \infty$ sequentially, the NTK $\Theta^{(L)}$ converges in probability to a deterministic limiting kernel:*

$$\Theta^{(L)} \to \Theta_\infty^{(L)} \otimes Id_{n_L}.$$

*The scalar kernel $\Theta_\infty^{(L)} : \mathbb{R}^{n_0} \times \mathbb{R}^{n_0} \to \mathbb{R}$ is defined recursively by*

$$\Theta_\infty^{(1)}(x, x') = \Sigma^{(1)}(x, x')$$
$$\Theta_\infty^{(L+1)}(x, x') = \Theta_\infty^{(L)}(x, x')\dot{\Sigma}^{(L+1)}(x, x') + \Sigma^{(L+1)}(x, x'),$$

*where*

$$\dot{\Sigma}^{(L+1)}(x, x') = \mathbb{E}_{f \sim \mathcal{N}(0,\Sigma^{(L)})} \left[ \dot{\sigma}(f(x)) \dot{\sigma}(f(x')) \right],$$

*taking the expectation with respect to a centered Gaussian process $f$ of covariance $\Sigma^{(L)}$, and where $\dot{\sigma}$ denotes the derivative of $\sigma$.*

*Proof.* The proof is again by induction. When $L = 1$, there is no hidden layer and therefore no limit to be taken. The neural tangent kernel is a sum over the entries of $W^{(0)}$ and those of $b^{(0)}$:

$$\Theta_{kk'}(x, x') = \frac{1}{n_0} \sum_{i=1}^{n_0} \sum_{j=1}^{n_1} x_i x_i' \delta_{jk} \delta_{jk'} + \beta^2 \sum_{j=1}^{n_1} \delta_{jk} \delta_{jk'}$$

$$= \frac{1}{n_0} x^T x' \delta_{kk'} + \beta^2 \delta_{kk'} = \Sigma^{(1)}(x, x')\delta_{kk'}.$$

Here again, the key to prove the induction step is the observation that a network of depth $L + 1$ is an $L$-network mapping the inputs $x$ to the preactivations of the $L$-th layer $\tilde{\alpha}^{(L)}(x)$ followed by a nonlinearity and a random affine function. For a network of depth $L + 1$, let us therefore split the parameters into the parameters $\tilde{\theta}$ of the first $L$ layers and those of the last layer $(W^{(L)}, b^{(L)})$.

By Proposition 1 and the induction hypothesis, as $n_1, ..., n_{L-1} \to \infty$ the pre-activations $\tilde{\alpha}_i^{(L)}$ are iid centered Gaussian with covariance $\Sigma^{(L)}$ and the neural tangent kernel $\Theta_{ii'}^{(L)}(x, x')$ of the smaller network converges to a deterministic limit:

$$\left(\partial_{\tilde{\theta}} \tilde{\alpha}_i^{(L)}(x; \theta)\right)^T \partial_{\tilde{\theta}} \tilde{\alpha}_{i'}^{(L)}(x'; \theta) \to \Theta_\infty^{(L)}(x, x') \delta_{ii'}.$$

We can split the neural tangent network into a sum over the parameters $\tilde{\theta}$ of the first $L$ layers and the remaining parameters $W^{(L)}$ and $b^{(L)}$.

For the first sum let us observe that by the chain rule:

$$\partial_{\tilde{\theta}_p} f_{\theta,k}(x) = \frac{1}{\sqrt{n_L}} \sum_{i=1}^{n_L} \partial_{\tilde{\theta}_p} \tilde{\alpha}_i^{(L)}(x; \theta) \dot{\sigma}(\tilde{\alpha}_i^{(L)}(x; \theta)) W_{ik}^{(L)}.$$

By the induction hypothesis, the contribution of the parameters $\tilde{\theta}$ to the neural tangent kernel $\Theta_{kk'}^{(L+1)}(x, x')$ therefore converges as $n_1, ..., n_{L-1} \to \infty$:

$$\frac{1}{n_L} \sum_{i,i'=1}^{n_L} \Theta_{ii'}^{(L)}(x, x') \dot{\sigma}\left(\tilde{\alpha}_i^{(L)}(x; \theta)\right) \dot{\sigma}\left(\tilde{\alpha}_{i'}^{(L)}(x'; \theta)\right) W_{ik}^{(L)} W_{i'k'}^{(L)}$$

$$\to \frac{1}{n_L} \sum_{i=1}^{n_L} \Theta_\infty^{(L)}(x, x') \dot{\sigma}\left(\tilde{\alpha}_i^{(L)}(x; \theta)\right) \dot{\sigma}\left(\tilde{\alpha}_i^{(L)}(x'; \theta)\right) W_{ik}^{(L)} W_{ik'}^{(L)}$$

By the law of large numbers, as $n_L \to \infty$, this tends to its expectation which is equal to

$$\Theta_\infty^{(L)}(x, x') \dot{\Sigma}^{(L+1)}(x, x') \delta_{kk'}.$$

It is then easy to see that the second part of the neural tangent kernel, the sum over $W^{(L)}$ and $b^{(L)}$ converges to $\Sigma^{(L+1)} \delta_{kk'}$ as $n_1, ..., n_L \to \infty$. $\square$

## A.2 Asymptotics during Training

Given a training direction $t \mapsto d_t \in \mathcal{F}$, a neural network is trained in the following manner: the parameters $\theta_p$ are initialized as iid $\mathcal{N}(0, 1)$ and follow the differential equation:

$$\partial_t \theta_p(t) = \left\langle \partial_{\theta_p} F^{(L)}, d_t \right\rangle_{p^{in}}.$$

In this context, in the infinite-width limit, the NTK stays constant during training:

**Theorem 2.** *Assume that $\sigma$ is a Lipschitz, twice differentiable nonlinearity function, with bounded second derivative. For any $T$ such that the integral $\int_0^T \|d_t\|_{p^{in}} dt$ stays stochastically bounded, as $n_1, ..., n_{L-1} \to \infty$ sequentially, we have, uniformly for $t \in [0, T]$,*

$$\Theta^{(L)}(t) \to \Theta_\infty^{(L)} \otimes Id_{n_L}.$$

*As a consequence, in this limit, the dynamics of $f_\theta$ is described by the differential equation*

$$\partial_t f_{\theta(t)} = \Phi_{\Theta_\infty^{(L)} \otimes Id_{n_L}} \left(\langle d_t, \cdot \rangle_{p^{in}}\right).$$

*Proof.* As in the previous theorem, the proof is by induction on the depth of the network. When $L = 1$, the neural tangent kernel does not depend on the parameters, it is therefore constant during training.

For the induction step, we again split an $L + 1$ network into a network of depth $L$ with parameters $\tilde{\theta}$ and top layer connection weights $W^{(L)}$ and bias $b^{(L)}$. The smaller network follows the training direction

$$d'_t = \dot{\sigma}\left(\tilde{\alpha}^{(L)}(t)\right) \left(\frac{1}{\sqrt{n_L}} W^{(L)}(t)\right)^T d_t$$

for $i = 1, \ldots, n_L$, where the function $\tilde{\alpha}_i^{(L)}(t)$ is defined as $\tilde{\alpha}_i^{(L)}(\cdot; \theta(t))$. We now want to apply the induction hypothesis to the smaller network. For this, we need to show that $\int_0^T \|d_t'\|_{p^{in}} \mathrm{d}t$ is stochastically bounded as $n_1, \ldots, n_L \to \infty$. Since $\sigma$ is a $c$-Lipschitz function, we have that

$$\|d_t'\|_{p^{in}} \le c\|\frac{1}{\sqrt{n_L}} W^{(L)}(t)\|_{op} \|d_t\|_{p^{in}}.$$

To apply the induction hypothesis, we now need to bound $\|\frac{1}{\sqrt{n_L}} W^{(L)}(t)\|_{op}$. For this, we use the following lemma, which is proven in Appendix A.3 below:

**Lemma 1.** *With the setting of Theorem 2, for a network of depth $L + 1$, for any $\ell = 1, \ldots, L$, we have the convergence in probability:*

$$\lim_{n_L \to \infty} \cdots \lim_{n_1 \to \infty} \sup_{t \in [0,T]} \|\frac{1}{\sqrt{n_\ell}} \left( W^{(\ell)}(t) - W^{(\ell)}(0) \right)\|_{op} = 0$$

From this lemma, to bound $\|\frac{1}{\sqrt{n_L}} W^{(L)}(t)\|_{op}$, it is hence enough to bound $\|\frac{1}{\sqrt{n_L}} W^{(L)}(0)\|_{op}$. From the law of large numbers, we obtain that the norm of each of the $n_{L+1}$ rows of $W^{(L)}(0)$ is bounded, and hence that $\|\frac{1}{\sqrt{n_L}} W^{(L)}(0)\|_{op}$ is bounded (keep in mind that $n_{L+1}$ is fixed, while $n_1, \ldots, n_L$ grow).

From the above considerations, we can apply the induction hypothesis to the smaller network, yielding, in the limit as $n_1, \ldots, n_L \to \infty$ (sequentially), that the dynamics is governed by the constant kernel $\Theta_\infty^{(L)}$:

$$\partial_t \tilde{\alpha}_i^{(L)}(t) = \frac{1}{\sqrt{n_L}} \Phi_{\Theta_\infty^{(L)}} \left( \left\langle \dot{\sigma}\left(\tilde{\alpha}_i^{(L)}(t)\right) \left(W_i^{(L)}(t)\right)^T d_t, \cdot \right\rangle_{p^{in}} \right).$$

At the same time, the parameters of the last layer evolve according to

$$\partial_t W_{ij}^{(L)}(t) = \frac{1}{\sqrt{n_L}} \left\langle \alpha_i^{(L)}(t), d_{t,j} \right\rangle_{p^{in}}.$$

We want to give an upper bound on the variation of the weights columns $W_i^{(L)}(t)$ and of the activations $\tilde{\alpha}_i^{(L)}(t)$ during training in terms of $L^2$-norm and $p^{in}$-norm respectively. Applying the Cauchy-Schwarz inequality for each $j$, summing and using $\partial_t \|\cdot\| \le \|\partial_t \cdot\|$), we have

$$\partial_t \left\| W_i^{(L)}(t) - W_i^{(L)}(0) \right\|_2 \le \frac{1}{\sqrt{n_L}} \|\alpha_i^{(L)}(t)\|_{p^{in}} \|d_t\|_{p^{in}}.$$

Now, observing that the operator norm of $\Phi_{\Theta_\infty^{(L)}}$ is equal to $\|\Theta_\infty^{(L)}\|_{op}$, defined in the introduction of Appendix A, and using the Cauchy-Schwarz inequality, we get

$$\partial_t \left\| \tilde{\alpha}_i^{(L)}(t) - \tilde{\alpha}_i^{(L)}(0) \right\|_{p^{in}} \le \frac{1}{\sqrt{n_L}} \left\| \Theta_\infty^{(L)} \right\|_{op} \left\| \dot{\sigma}\left(\tilde{\alpha}_i^{(L)}(t)\right) \right\|_\infty \left\| W_i^{(L)}(t) \right\|_2 \|d_t\|_{p^{in}},$$

where the sup norm $\|\cdot\|_\infty$ is defined by $\|f\|_\infty = \sup_x |f(x)|$.

To bound both quantities simultaneously, study the derivative of the quantity

$$A(t) = \|\alpha_i^{(L)}(0)\|_{p^{in}} + c \left\| \tilde{\alpha}_i^{(L)}(t) - \tilde{\alpha}_i^{(L)}(0) \right\|_{p^{in}} + \|W_i^{(L)}(0)\|_2 + \left\| W_i^{(L)}(t) - W_i^{(L)}(0) \right\|_2.$$

We have

$$\partial_t A(t) \le \frac{1}{\sqrt{n_L}} \left( c^2 \left\| \Theta_\infty^{(L)} \right\|_{op} \left\| W_i^{(L)}(t) \right\|_2 + \|\alpha_i^{(L)}(t)\|_{p^{in}} \right) \|d_t\|_{p^{in}}$$

$$\le \frac{\max\{c^2 \|\Theta_\infty^{(L)}\|_{op}, 1\}}{\sqrt{n_L}} \|d_t\|_{p^{in}} A(t),$$

where, in the first inequality, we have used that $|\dot{\sigma}| \leq c$ and, in the second inequality, that the sum $\|W_i^{(L)}(t)\|_2 + ||\alpha_i^{(L)}(t)||_{p^{in}}$ is bounded by $A(t)$. Applying Grönwall's Lemma, we now get

$$A(t) \leq A(0) \exp\left(\frac{\max\{c^2\|\Theta_\infty^{(L)}\|_{op}, 1\}}{\sqrt{n_L}} \int_0^t \|d_s\|_{p^{in}} ds\right).$$

Note that $\|\Theta_\infty^{(L)}\|_{op}$ is constant during training. Clearly the value inside of the exponential converges to zero in probability as $n_L \to \infty$ given that the integral $\int_0^t \|d_t\|_{p^{in}} ds$ stays stochastically bounded. The variations of the activations $\left\|\tilde{\alpha}_i^{(L)}(t) - \tilde{\alpha}_i^{(L)}(0)\right\|_{p^{in}}$ and weights $\left\|W_i^{(L)}(t) - W_i^{(L)}(0)\right\|_2$ are bounded by $c^{-1}(A(t) - A(0))$ and $A(t) - A(0)$ respectively, which converge to zero at rate $O\left(\frac{1}{\sqrt{n_L}}\right)$.

We can now use these bounds to control the variation of the NTK and to prove the theorem. To understand how the NTK evolves, we study the evolution of the derivatives with respect to the parameters. The derivatives with respect to the bias parameters of the top layer $\partial_{b_j^{(L)}} f_{\theta,j'}$ are always equal to $\delta_{jj'}$. The derivatives with respect to the connection weights of the top layer are given by

$$\partial_{W_{ij}^{(L)}} f_{\theta,j'}(x) = \frac{1}{\sqrt{n_L}} \alpha_i^{(L)}(x;\theta)\delta_{jj'}.$$

The pre-activations $\tilde{\alpha}_i^{(L)}$ evolve at a rate of $\frac{1}{\sqrt{n_L}}$ and so do the activations $\alpha_i^{(L)}$. The summands $\partial_{W_{ij}^{(L)}} f_{\theta,j'}(x) \otimes \partial_{W_{ij}^{(L)}} f_{\theta,j''}(x')$ of the NTK hence vary at rate of $n_L^{-3/2}$ which induces a variation of the NTK of rate $\frac{1}{\sqrt{n_L}}$.

Finally let us study the derivatives with respect to the parameters of the lower layers

$$\partial_{\tilde{\theta}_k} f_{\theta,j}(x) = \frac{1}{\sqrt{n_L}} \sum_{i=1}^{n_L} \partial_{\tilde{\theta}_k} \tilde{\alpha}_i^{(L)}(x;\theta)\dot{\sigma}\left(\tilde{\alpha}_i^{(L)}(x;\theta)\right) W_{ij}^{(L)}.$$

Their contribution to the NTK $\Theta_{jj'}^{(L+1)}(x,x')$ is

$$\frac{1}{n_L} \sum_{i,i'=1}^{n_L} \Theta_{ii'}^{(L)}(x,x')\dot{\sigma}\left(\tilde{\alpha}_i^{(L)}(x;\theta)\right) \dot{\sigma}\left(\tilde{\alpha}_{i'}^{(L)}(x';\theta)\right) W_{ij}^{(L)}W_{i'j'}^{(L)}.$$

By the induction hypothesis, the NTK of the smaller network $\Theta^{(L)}$ tends to $\Theta_\infty^{(L)}\delta_{ii'}$ as $n_1, ..., n_{L-1} \to \infty$. The contribution therefore becomes

$$\frac{1}{n_L} \sum_{i=1}^{n_L} \Theta_\infty^{(L)}(x,x')\dot{\sigma}\left(\tilde{\alpha}_i^{(L)}(x;\theta)\right) \dot{\sigma}\left(\tilde{\alpha}_i^{(L)}(x';\theta)\right) W_{ij}^{(L)}W_{ij'}^{(L)}.$$

The connection weights $W_{ij}^{(L)}$ vary at rate $\frac{1}{\sqrt{n_L}}$, inducing a change of the same rate to the whole sum. We simply have to prove that the values $\dot{\sigma}(\tilde{\alpha}_i^{(L)}(x;\theta))$ also change at rate $\frac{1}{\sqrt{n_L}}$. Since the second derivative of $\sigma$ is bounded, we have that

$$\partial_t\left(\dot{\sigma}\left(\tilde{\alpha}_i^{(L)}(x;\theta(t))\right)\right) = O\left(\partial_t\tilde{\alpha}_i^{(L)}(x;\theta(t))\right).$$

Since $\partial_t\tilde{\alpha}_i^{(L)}(x;\theta(t))$ goes to zero at a rate $\frac{1}{\sqrt{n_L}}$ by the bound on $A(t)$ above, this concludes the proof.

$\square$

It is somewhat counterintuitive that the variation of the activations of the hidden layers $\alpha_i^{(\ell)}$ during training goes to zero as the width becomes large[1]. It is generally assumed that the purpose of the activations of the hidden layers is to learn "good" representations of the data during training. However note that even though the variation of each individual activation shrinks, the number of neurons grows, resulting in a significant collective effect. This explains why the training of the parameters of each layer $\ell$ has an influence on the network function $f_\theta$ even though it has asymptotically no influence on the individual activations of the layers $\ell'$ for $\ell < \ell' < L$.

### A.3 A Priori Control during Training

The goal of this section is to prove Lemma 1, which is a key ingredient in the proof of Theorem 2. Let us first recall it:

**Lemma 1.** *With the setting of Theorem 2, for a network of depth $L + 1$, for any $\ell = 1, \ldots, L$, we have the convergence in probability:*

$$\lim_{n_L \to \infty} \cdots \lim_{n_1 \to \infty} \sup_{t \in [0,T]} \| \frac{1}{\sqrt{n_\ell}} \left( W^{(\ell)}(t) - W^{(\ell)}(0) \right) \|_{op} = 0$$

*Proof.* We prove the lemma for all $\ell = 1, \ldots, L$ simultaneously, by expressing the variation of the weights $\frac{1}{\sqrt{n_\ell}} W^{(\ell)}$ and activations $\frac{1}{\sqrt{n_\ell}} \tilde{\alpha}^{(\ell)}$ in terms of 'back-propagated' training directions $d^{(1)}, \ldots, d^{(L)}$ associated with the lower layers and the NTKs of the corresponding subnetworks:

1. At all times, the evolution of the preactivations and weights is given by:

$$\partial_t \tilde{\alpha}^{(\ell)} = \Phi_{\Theta^{(\ell)}} \left( < d_t^{(\ell)}, \cdot >_{p^{in}} \right)$$

$$\partial_t W^{(\ell)} = \frac{1}{\sqrt{n_\ell}} < \alpha^{(\ell)}, d_t^{(\ell+1)} >_{p^{in}},$$

where the layer-wise training directions $d^{(1)}, \ldots, d^{(L)}$ are defined recursively by

$$d_t^{(\ell)} = \begin{cases} d_t & \text{if } \ell = L + 1 \\ \dot{\sigma} \left( \tilde{\alpha}^{(\ell)} \right) \left( \frac{1}{\sqrt{n_\ell}} W^{(\ell)} \right)^T d_t^{(\ell+1)} & \text{if } \ell \leq L, \end{cases}$$

and where the sub-network NTKs $\Theta^{(\ell)}$ satisfy

$$\Theta^{(1)} = \left[ \left[ \frac{1}{\sqrt{n_0}} \alpha^{(0)} \right]^T \left[ \frac{1}{\sqrt{n_0}} \alpha^{(0)} \right] \right] \otimes Id_{n_\ell} + \beta^2 \otimes Id_{n_\ell}$$

$$\Theta^{(\ell+1)} = \frac{1}{\sqrt{n_\ell}} W^{(\ell)} \dot{\sigma}(\tilde{\alpha}^{(\ell)}) \Theta^{(\ell)} \dot{\sigma}(\tilde{\alpha}^{(\ell)}) \frac{1}{\sqrt{n_\ell}} W^{(\ell)}$$

$$+ \left[ \left[ \frac{1}{\sqrt{n_\ell}} \alpha^{(\ell)} \right]^T \left[ \frac{1}{\sqrt{n_\ell}} \alpha^{(\ell)} \right] \right] \otimes Id_{n_\ell} + \beta^2 \otimes Id_{n_\ell}.$$

2. Set $w^{(k)}(t) := \left\| \frac{1}{\sqrt{n_k}} W^{(k)}(t) \right\|_{op}$ and $a^{(k)}(t) := \left\| \frac{1}{\sqrt{n_k}} \alpha^{(k)}(t) \right\|_{p^{in}}$. The identities of the previous step yield the following recursive bounds:

$$\left\| d_t^{(\ell)} \right\|_{p^{in}} \leq c w^{(\ell)}(t) \left\| d_t^{(\ell+1)} \right\|_{p^{in}},$$

where $c$ is the Lipschitz constant of $\sigma$. These bounds lead to

$$\left\| d_t^{(\ell)} \right\|_{p^{in}} \leq c^{L+1-\ell} \prod_{k=\ell}^{L} w^{(k)}(t) \left\| d_t \right\|_{p^{in}}.$$

For the subnetworks NTKs we have the recursive bounds

$$\| \Theta^{(1)} \|_{op} \leq (a^{(0)}(t))^2 + \beta^2.$$

$$\| \Theta^{(\ell+1)} \|_{op} \leq c^2 (w^{(\ell)}(t))^2 \| \Theta^{(\ell)} \|_{op} + (a^{(\ell)}(t))^2 + \beta^2,$$

which lead to

$$\| \Theta^{(\ell+1)} \|_{op} \leq \mathcal{P} \left( a^{(1)}, \ldots, a^{(\ell)}, w^{(1)}, \ldots, w^{(\ell)} \right),$$

where $\mathcal{P}$ is a polynomial which only depends on $\ell, c, \beta$ and $p^{in}$.

3. Set

$$\tilde{a}^{(k)}(t) := \left\|\frac{1}{\sqrt{n_k}}\left(\tilde{\alpha}^{(k)}(t) - \tilde{\alpha}^{(k)}(0)\right)\right\|_{p^{in}}$$

$$\tilde{w}^{(k)}(t) := \left\|\frac{1}{\sqrt{n_k}}\left(W^{(k)}(t) - W^{(k)}(0)\right)\right\|_{op}$$

and define

$$A(t) = \sum_{k=1}^{L} a^{(k)}(0) + c\tilde{a}^{(k)}(t) + w^{(k)}(0) + \tilde{w}^{(k)}(t).$$

Since $a^{(k)}(t) \le a^{(k)}(0) + c\tilde{a}^{(k)}(t)$ and $w^{(k)}(t) \le w^{(k)}(0) + \tilde{w}^{(k)}(t)$, controlling $A(t)$ will enable us to control the $a^{(k)}(t)$ and $w^{(k)}(t)$. Using the formula at the beginning of the first step, we obtain

$$\partial_t \tilde{a}^{(\ell)}(t) \le \frac{1}{\sqrt{n_\ell}}\|\Theta^{(\ell)}(t)\|_{op}\|d_t^{(\ell)}\|_{p^{in}}$$

$$\partial_t \tilde{w}^{(\ell)}(t) \le \frac{1}{\sqrt{n_\ell}}a^{(\ell)}(t)\,\|d_t^{(\ell+1)}\|_{p^{in}}.$$

This allows one to bound the derivative of $A(t)$ as follows:

$$\partial_t A(t) \le \sum_{\ell=1}^{L} \frac{c}{\sqrt{n_\ell}}\|\Theta^{(\ell)}(t)\|_{op}\|d_t^{(\ell)}\|_{p^{in}} + \frac{1}{\sqrt{n_\ell}}a^{(\ell)}(t)\,\|d_t^{(\ell+1)}\|_{p^{in}}.$$

Using the polynomial bounds on $\|\Theta^{(\ell)}(t)\|_{op}$ and $\|d_t^{(\ell+1)}\|_{p^{in}}$ in terms of the $a^{(k)}$ and $w^{(k)}$ for $k = 1, \ldots \ell$ obtained in the previous step, we get that

$$\partial_t A(t) \le \frac{1}{\sqrt{\min\{n_1, \ldots, n_L\}}} \mathcal{Q}\left(w^{(1)}(t), \ldots, w^{(L)}(t), a^{(1)}(t), \ldots, a^{(L)}(t)\right)\|d_t\|_{p^{in}},$$

where the polynomial $Q$ only depends on $L, c, \beta$ and $p^{in}$ and has positive coefficients. As a result, we can use $a^{(k)}(t) \le a^{(k)}(0) + c\tilde{a}^{(k)}(t)$ and $w^{(k)}(t) \le w^{(k)}(0) + \tilde{w}^{(k)}(t)$ to get the polynomial bound

$$\partial_t A(t) \le \frac{1}{\sqrt{\min\{n_1, \ldots, n_L\}}} \tilde{\mathcal{Q}}(A(t))\,\|d_t\|_{p^{in}}.$$

4. Let us now observe that $A(0)$ is stochastically bounded as we take the sequential limit $\lim_{n_L \to \infty} \cdots \lim_{n_1 \to \infty}$ as in the statement of the lemma. In this limit, we indeed have that $w^{(\ell)}$ and $a^{(\ell)}$ are convergent: we have $w^{(\ell)} \to 0$, while $a^{(\ell)}$ converges by Proposition 1.

The polynomial control we obtained on the derivative of $A(t)$ now allows one to use (a nonlinear form of, see e.g. (5)) Grönwall's Lemma: we obtain that $A(t)$ stays uniformly bounded on $[0, \tau]$ for some $\tau = \tau(n_1, \ldots, n_L) > 0$, and that $\tau \to T$ as $\min(n_1, \ldots, n_L) \to \infty$, owing to the $\frac{1}{\sqrt{\min\{1,\ldots,n_L\}}}$ in front of the polynomial. Since $A(t)$ is bounded, the differential bound on $A(t)$ gives that the derivative $\partial_t A(t)$ converges uniformly to 0 on $[0, \tau]$ for any $\tau < T$, and hence $A(t) \to A(0)$. This concludes the proof of the lemma.

$\square$

## Footnotes

[1] As a consequence, the pre-activations stay Gaussian during training as well, with the same covariance $\Sigma^{(\ell)}$.