[Reviews · NeurIPS 2018]

Reviewer 1



The authors prove that networks of infinite width trained with SGD and (infinitely) small step size evolve according to a differential equation, the solution of which depends only on the covariance kernel of the data and, in the case of L2 regression, on the eigenspectrum of the Kernel. I believe this is a breakthrough result in the field of neural network theory. It elevates the analysis of infinitely wide networks from the study of the static initial function to closely predicting the entire training path. There are a plethora of powerful consequences about infinitely wide, fully-connected networks: - They cannot learn information not contained in the covariance matrix - Change to latent representation and parameters tends to zero as width goes to infinity. - Therefore, disentangled intermediate representations are not learnt - Because the weight change converges to zero, effective depth goes to 2. - The complexity of nonlinearity design, including placing different nonlinearities in different layers, is reduced to NTK design. - If all inputs have the same length, the NTK is a 1d function. Therefore choosing nonlinearities in all layers reduces to choosing a single 1d function. - If all inputs have the same length, assuming a 1-to-1 mapping between NTL and sigma, all networks are equivalent to single-hidden-layer networks. This doesn't even mention the new analytical tools introduced that will no doubt lead to further important discoveries. There are some notation / correctness issues. Consider the term "Gaussian function $f$ of covariance $\Sigma^{(L)}$". In line 122, $f$ is written as a function of a single variable, i.e. $f(x)$ and $f(x')$. But if $f$ is a Gaussian function of one variable, it only has a variance, no covariance. Hence, the statement "$f$ of covariance $\Sigma^{(L)}$" makes no sense. Further, also on line 122, we have the expression $\mathbb{E}_f$. But if $f$ is a Gaussian function, it is not a random variable, and therefore we cannot take an expectation over it. Finally, in line 126, we have $f \sim \mathcal{N}(0,\Sigma^{(L-1)})$. Again, this notation suggests that $f$ is a random variable, not a Gaussian function. The notation $\mathcal{N}(0,\Sigma^{(L-1)})$ also suggests that the scalar $\Sigma^{(L-1)}$ is the variance of a 1d random variable, NOT the covariance, as is claimed earlier. The larger issue is that the notation chosen by the authors suggests that the covariance $\Sigma^{(L)}$ in a given network is only a function of the covariance in the previous layer, $\Sigma^{(L-1)}$. This is not the case. It depends on $\Sigma^{(L-1)}$ and the variance of $\tilde{\alpha}(x)$ in layer L-1 and the variance of $\tilde{\alpha}(x')$ in layer L-1. In fact, one cannot define the covariance $\Sigma^{(L)}$ recursively in isolation. One has to resursively define the triple (variance(x),variance(x'),covariance(x,x')) jointly via recursion. The variances then also show up in the definition of the NTK and pretty much everywhere also the covariance shows up. The authors should fix this ambiguity in their final version. Some other minor comments: - line 63: $\alpha^{(l)}$ is missing the $^$ sign. - in the beginning, $L$ is used to denote the depth of the network and $\ell$ is used for layer recursion. But halfway through, $L$ starts being used for layer recursion. It might be worth changing this for consistency. - in remark 1, it is claimed that under Xavier initialization, weight matrix entries are initialized to independent Gaussians of variance $\frac{1}{n_l}$. In fact, Xavier initialization uses independent centered uniform random variables of variance $\frac{2}{n_l + n_{l+1}}$. Variance $\frac{1}{n_l}$ is sometimes called LeCun initialization. Further, in the paper that introduced Xavier initialization, biases were initialized to zero, not $\mathcal{N}(0,1)$. In fact, initializing biases as $\mathcal{N}(0,1)$ is a terrible choices that is not used, as it would replace a substantial portion of the signal variance with bias in each layer. ##### Rebuttal response #### My misunderstanding regarding the covariance / variance issue probably resulted from me having little experience with the common terminology / notation of Kernel machines. I interpreted the recursive formula as trying to establish a recursion for specific values of x,x', i.e. recursing \Sigma^L(x,x') based on the value of \Sigma^L-1(x,x'). However, I see now that you are trying to recurse the entire function \Sigma^L(x,x') as defined for all pairs x,x' based on the entire function of \Sigma^L-1(x,x') for all pairs x,x'. ... That makes sense. Again, nice paper :)

Reviewer 2



Summary: The authors show that an artificial neural network (ANN) with infinite width can be interpreted using Neural Tangent Kernels (NTK). Using the NTK, they show that ANN converges faster on a few large principal components of the NTK which they argue is a motivation for early stopping. Quality: Quite frankly I had a hard time understanding all the mathematical definitions related to kernels. I cannot do better than make an educated guess. Clarity: The motivation, theoretical explanations and experiments were well explained in general. Unfortunately I could not understand it in details. Originality: I am not familiar enough with the subject to evaluate the originality. Significance: The final results and assumptions related to skewed distribution of eigenvalues of the input data with respect to the NTK seems related to similar observations made with respect to the Hessian [1]. Due to my lack of understanding of the theory behind the NTK I cannot tell whether those are truly related or not. [1] Karakida, Ryo, Shotaro Akaho, and Shun-ichi Amari. "Universal Statistics of Fisher Information in Deep Neural Networks: Mean Field Approach." arXiv preprint arXiv:1806.01316 (2018).

Reviewer 3



Update: I am getting a better understanding of this paper after a second look, with the help of the rebuttal and the discussions. I think this is a good paper and am changing the evaluation accordingly. The phenomenon this paper pointed out is rather surprising. In particular in the square loss case, it says gradient descent acts like a linear regression, in which all bust the last layer barely moves, and the last layer tries to learn the best linear combination of the (almost fixed) hidden representation. I did some experiments and find that this phenomenon is indeed true -- fixing a time horizon T and letting the dimension of intermediate hidden layers be large, the last layer changes more significantly than all other layers. This is really interesting and maybe the authors can bring this fact in a stronger tone. About the relationship with Mei et al., I agree with the rebuttal -- the views are largely complementary. While both focuses on wide neural networks, Mei et al. identifies the exact dynamics for one-hidden-layer nets with a fixed size, while this work assumes the size goes to infinity but is able to characterize arbitrary deep nets. ---------------------- This paper proposes a Neural Tangent Kernel, a model of the gradient descent dynamics of neural networks in the function space. In the case of linear parametrization, this is a kernel gradient descent with a constant kernel (the tangent kernel). In the non-linear case, this result cease to hold, but this paper shows that the neural tangent kernel converges, in the infinite width limit, to a fixed kernel over any finite time horizon. Consequently, the training dynamics of a very wide neural net can be similar to that of a least-squares when it uses the square loss. I feel like this is a solid piece of theoretical work but am not quite sure about its connection to (and significant among) concurrent results. In particular, does the limiting invariant kernel say something similar to "no spurious local minima" or "no bad critical point / essential convexity" in the infinite-n limit? One possibility is to compare with Mei et al. (2018), which builds on the observation that one-hidden-layer net is a linear function of the empirical distributions of weights, and thereby the square loss is convex. This empirical distribution is a good "function space" for one-hidden-layer nets. As both papers shows some sort of "linear regression"-like scenario in the infinite-width limit, I wonder if any interesting interpretations / connections can pop up from here. I also find it hard to pin down some technical details about kernels, especially Section 3.0 about functional derivatives and kernel gradients (perhaps not easy anyway due to the space constraint). But it would still be great if this part can be made more approachable to non-kernel-experts. Reference: Mei, Song, Andrea Montanari, and Phan-Minh Nguyen. "A Mean Field View of the Landscape of Two-Layers Neural Networks." arXiv preprint arXiv:1804.06561 (2018).